# Coping Strategies as a Moderator for the Association between Intimate Partner Violence and Depression and Anxiety Symptoms among Transgender Women

**DOI:** 10.3390/ijerph20115927

**Published:** 2023-05-23

**Authors:** Shannon S. Gray, Kayla Marie Sizemore, H. Jonathon Rendina

**Affiliations:** 1Institute for Health, Healthcare Policy, and Aging Research, Rutgers University, 112 Paterson St., New Brunswick, NJ 08901, USA; msizemore@ifh.rutgers.edu; 2Department of Psychiatry, Robert Wood Johnson Medical School, Rutgers University, 125 Paterson Street, New Brunswick, NJ 08901, USA; 3Department of Epidemiology, Milken Institute School of Public Health, George Washington University, 950 New Hampshire Ave, NW, Washington, DC 20052, USA; jrendina@gwu.edu; 4Whitman-Walker Institute, 1377 R St., NW, Suite 200, Washington, DC 20009, USA

**Keywords:** adaptive coping, intimate partner violence, anxiety, depression, transgender women, transgender health, minority stress

## Abstract

Evidence suggests that intimate partner violence (IPV) is associated with negative mental health outcomes. There is currently limited research on the impact of IPV on the outcomes of mental health for transgender women. The current study aimed to examine the relationship between intimate partner violence, coping skills, depression, and anxiety in a sample of transgender women. Hierarchical regression analyses were conducted examining the relationship of IPV and depression and anxiety symptoms, where coping skills moderate this relationship. The results suggest that those with experiences of IPV are more likely to have symptoms of depression and anxiety. For individuals with no experiences of IPV and low depression, high levels of emotional processing coping and acceptance coping buffered this relationship. For individuals with more experiences of IPV and more depressive symptoms, coping skills did not show to buffer this relationship. These same coping skills did not show evidence for buffering anxiety symptoms for transgender women with low or high levels of IPV. The results, implications, and limitations of this study and suggestions for further research are discussed.

## 1. Introduction

While there has been much progress for the rights and visibility of transgender and gender-diverse (TGD) populations [1,2], substantial discrimination and marginalization continue to occur [1,2,3,4,5,6]. Specifically, studies have shown that transgender women, in particular, experience high rates of marginalization, discrimination, and violence, compared to the U.S. general population, across multiple areas of life [7,8,9], for example, in healthcare settings [7,10,11,12,13], in the workplace [7,9], in encounters with law enforcement [7,9,14,15,16], and in family and romantic relationships [7,8,9,17]. Intimate partner violence (IPV), in particular, is one of the most reported forms of violence among transgender women [7,8,9,14,17,18,19,20,21].

IPV includes intimate partner abuse (IPA), domestic violence, and dating violence and occurs when one partner physically, emotionally, psychologically, or financially abuses their intimate partner. There are multiple risk factors which contribute to the high rates of IPV experienced by transgender women specifically [7,8,22]. Substance use [23] and sex work, for instance [7,24], have been evidenced to place transgender women, specifically, at increased risk for experiencing IPV and sexual violence [7,8,9,20,24,25]. Additionally, housing insecurity and poverty rates are high within the transgender female population and have been associated with an increased risk of experiencing financial-related IPV [7,9]. Additionally, a history of childhood abuse has also been reported at high rates within the transgender population [7,8,9,20,25], and the World Health Organization (WHO) reports that childhood abuse is a strong risk factor toward experiencing IPV later in life [26]. Other risk factors which can contribute to experiences of IPV for transgender women are experiences of discrimination [6,7,24,27], familial abandonment [17,28], or lack of social support due to gender identity [9].

For transgender women, IPV is associated with increased mental health disparities such as anxiety, depression, posttraumatic stress disorder (PTSD), suicidal ideation [7,8,20], and an increased risk of death due to fatal violence [7,8,9,20]. For instance, researchers examining the effects of IPV on TGD individuals’ mental health found significant associations between anxiety and four types of IPV, as well as depression and three types of IPV. These findings suggest that multiple forms of IPV may increase TGD individuals’ levels of anxiety and depression [29]. In addition, among a sample of transgender women who reported experiencing more than one potentially traumatic event (PTE), 39% reported they had experienced IPV, 9% reported they had been a victim of sexual assault, and 27% reported they had been a victim of childhood sexual abuse. Further, 43% reported they had been threatened with death or bodily harm [30]. Further, in 2016, the National Coalition of Anti-Violence Programs reported 15 homicides specifically due to IPV within the lesbian, gay, bisexual, and transgender (LGBT) population. Of these homicides, 11% were transgender women and non-binary-identifying individuals. Fourteen of the individuals were killed by a current or past intimate partner [9]. This number of reported deaths is also speculated to be under-reported due to a history of census reporting excluding transgender individuals, and due to police and media outlets reporting inaccurate gender identification [7,9,31].

Important factors shown to be protective against negative mental health outcomes for TGD populations are social support and community connectedness, and the use of adaptive or facilitative coping methods [32,33,34,35,36,37]. Specifically, social support has been evidenced to decrease distress among transgender populations and act as a protective factor against suicidal ideation [32,33]. Further, facilitative and adaptive coping strategies (e.g., providing education, utilizing resources, and confrontation) have been associated with fewer symptoms of depression than avoidant or detachment-oriented coping (e.g., alcohol use and isolation from others) [32,38]. However, much of the literature has focused on coping strategies related to minority stress, discriminatory experiences, and not IPV-specific experiences among TGD individuals [38].

In this study we examined the relationship of IPV and depressive and anxiety symptoms. We assessed the potential moderating effect of four coping skills (e.g., acceptance, emotional processing, emotional expression, and social support) on the relationships between IPV, specifically, and depressive symptoms and anxiety symptoms. These coping skills were chosen based on prior research regarding coping methods shown to be protective against negative mental health outcomes for TGD populations [32,33,34,35,36,37]. It was hypothesized that depression and anxiety levels would be positively correlated with intimate partner violence, where more experiences of IPV would predict higher levels of depression and anxiety symptomatology. We hypothesized an interaction would be observed between IPV and each coping skill. Specifically, we wanted to examine whether the coping skills would moderate the association between experiences of IPV and anxious or depressive symptomatology.

## 2. Materials and Methods

### 2.1. Participants

Data were retrieved from a previous study, using the data from the baseline assessment [39]. The participants in this study included 212 transgender women. Inclusion criteria were that participants needed to be 18 years or older; identify as a transgender woman or a woman of transgender experience (i.e., assigned male sex at birth and currently identify as female); reside in the New York City metropolitan area; be able to complete a survey in English or with Spanish language translation assistance; be able to provide contact information; report ≥1 act of anal or vaginal sex in the past 60 days; and also report ≥1 day of illicit drug use in the past 60 days. Exclusion criteria included instances where the participant was unstable; reported serious psychiatric symptoms; reported current suicidal or homicidal ideation; presented evidence of gross cognitive impairment; was enrolled in a drug abuse treatment; was enrolled in an HIV risk intervention study; or was enrolled in a drug use intervention study; participant was unable to communicate in English.

The average age of participants was 34 years old. With regard to ethnicity, 31.6% of the participants were Black, 28.3% Latina, 24.5% White, and 15.6% other ethnicities. Just under half of participants reported their sexual orientation as straight (47.6%), followed by bisexual (18.4%) and queer (14.6%), gay (13.7%), and (5.6%) lesbian. Most participants (about 80%) had an income of USD 20,000 or below, and 73.6% of individuals reported being unemployed. Almost half of the participants (41.5%) reported an education level of high school or less, while just 29.2% reported receiving some college education. Additionally, 34.4% of the participants reported their HIV status as positive. Further, 64.2% of participants reported having had sex with someone as a means of income. Almost all participants (87.7%) reported having experienced at least one form of discrimination. Half of the participants reported being in a relationship (49.1%), and 65.1% reported having experienced at least one form of IPV in the past 5 years. The sample overall reported 51.7% clinically significant levels of anxiety on the BSI scale using cutoff scores according to Schulte-van Maaren et al. (2012) [40], and moderate to high levels of depression (61.3%) according to CES-D cutoff scores [41].

### 2.2. Measures

#### 2.2.1. Intimate Partner Violence

IPV was measured via a modified version of the Conflict Tactics Scale (CTS2) (Straus [42,43]. The CTS2 collects data on physical assault within a partnership (Straus [42,44]. In the current study, the measure contained 12 items which assessed an individual’s experiences of IPV over the previous five-year period. Participants answered each item with a “yes” or “no” response. Scores were calculated as a sum where higher scores indicated a greater number of experiences of IPV. The reliability coefficient for the scale, 𝛼 = 0.92, falls within the similar range obtained in previous studies [40], indicating confidence in the reliability of the measure.

#### 2.2.2. The Modified COPE

The Modified COPE [39] contains 40 items across 11 scales. Four coping skill subscales were examined: acceptance scale (4 items), emotional processing scale (4 items), emotional expression scale (4 items), and the social support scale (4 items). A sample item from the acceptance subscale was “I get used to the idea that it happened”. A sample item from the emotional processing subscale was “I delve into my feelings to get a thorough understanding of them”. A sample item from the emotional expression subscale was “I let my feelings come out freely”. A sample item from the social support subscale was “I get help and advice from other people”. Participants rated responses using a four-point Likert scale. Response options ranges from 1 (I usually don’t do this at all) to 4 (I usually do this a lot). An average of each participant’s responses for each subscale was calculated, with higher scores meaning greater use of that coping skills. The reliability coefficients for the scales were 0.78 for the acceptance scale, 0.81 for the emotional processing scale, 0.87 for the emotional expression scale, and 0.82 for the social support scale. All coefficients were similar to those obtained in previous studies [45,46,47], suggesting reliability and internal consistency.

#### 2.2.3. Center for Epidemiological Studies-Depression Scale

Symptoms of depression were measured via the Center for Epidemiological Studies- Depression scale (CES–D) [41]. The CES–D is utilized to assess the prevalence and severity of clinical depressive symptoms individuals may be experiencing. The CES–D contained 20 Items. Each item asked the participant to respond how often they felt each item in the past three-month period. A sample item from this scale was “I felt that I could not shake off the blues even with help from my family or friends”. Participants responded using a 4-point Likert scale. Response options ranged between 1 (rarely or none of the time) and 4 (most or all of the time). The reliability coefficient for the scale was 0.80, which is similar to that obtained in previous studies [48].

#### 2.2.4. The Brief Symptom Inventory

Anxiety symptoms were measured via the anxiety subscale of the Brief Symptom Inventory (BSI) [49]. The BSI measures self-reported clinical psychological symptoms across 9 symptom dimensions: somatization, obsession-compulsion, interpersonal sensitivity, depression, anxiety, hostility, phobic anxiety, paranoid ideation, and psychoticism. Participants in the current study only responded to the Anxiety dimension of the BSI. This dimension assessed clinical anxiety symptoms individuals might be experiencing over the past week. A sample item from this scale was “Feeling so restless you couldn’t sit well.”. Each item was answered using a 5-point Likert scale. Response options ranged between 0 (not at all) and 4 (extremely) The reliability coefficient for this scale was similar to that obtained in previous studies, α = 0.91 [40].

### 2.3. Analyses

Before conducting analyses, we mean centered all four coping subscales, and we calculated four interaction terms for each of the coping subscales and IPV. Descriptive statistics (means and standard deviations) and frequencies were examined to assess normality and skewness of each variable. Cronbach’s alpha was calculated for each scale to assess reliability as some measurement scales were modified from their original versions. Collinearity diagnostics were conducted to ensure each predictor’s contribution was unique by examining tolerance and the variation inflation factor (VIF) for each variable. A power analysis using Green’s recommendations (1991) [50] was conducted to ensure optimal sample size. We visually inspected the standardized residuals via scatterplots to evaluate linearity, normality, and homoscedasticity. Bivariate analyses were conducted to assess intercorrelation between variables.

Finally, we conducted four 3-step hierarchical linear regression analyses, examining outcomes for depression and again for anxiety outcomes using IBM SPSS v.25. Step one of the hierarchical regression analyses consisted of all control variables, including standard demographics and variables evidenced to impact the transgender female population specifically. Some of these included sex work, discrimination, and HIV status [6,7,8,9]. Step two introduced both the coping subscale and IPV to assess the relationship of each on the dependent variable. Step three included the interaction of the coping subscale and IPV to assess whether each coping subscale may provide a buffering effect for the relationship between IPV and each outcome (i.e., depressive and anxiety symptoms). Eight total regressions were conducted. We then plotted the interactions using Excel v.16.18.

## 3. Results

Descriptive statistics (means and standard deviations) are presented in Table 1. Optimal sample size analyses according to Green’s recommendations (1991) [50] resulted in a suggested sample size of 74–107 participants. We concluded that our sample size was sufficient. Residual and scatterplots were all randomly dispersed around zero, suggesting linearity, normality, and homoscedasticity. An examination of the Mahalanobis distance scores indicated no multivariate outliers. Pearson correlations resulted in nine statistically significant correlation coefficients, two of which were negative (see Table 2). However, no tolerance values were less than 0.10, and no VIF was greater than 10. Therefore, no significant collinearity was found, and assumptions for multicollinearity were met.

### 3.1. IPV and Coping Regression Models for Depression Outcomes

Table 3—At step one for each model, control variables accounted for 19.9% of the variation in depression outcomes. While adjusting for all other variables, age (β = −0.155, *p* = 0.022) and discrimination (β = 0.350, *p* = 0.000) were significantly associated with depression score.

#### 3.1.1. Acceptance Coping Model for Depression Outcomes

In the second step of the model for acceptance coping, IPV and acceptance coping were added. Age (β = −0.159, *p* = 0.018), discrimination (β = 0.300, *p* = 0.000), and IPV (β = 0.146, *p* = 0.046) were significantly associated with depression outcomes. The addition of these variables accounted for an additional 1.8% of variance in depression outcomes. In step three, the interaction term for IPV and acceptance coping was added into the model. Age (β = −0.146, *p* = 0.028), discrimination (β = 0.293, *p* = 0.000), IPV (β = 0.143, *p* = 0.047), acceptance coping (β = −0.262, *p* = 0.010), and the interaction variable (β = 0.266, *p* = 0.009) were significantly associated with depression outcomes. The addition of this variable accounted for an additional 2.7% of variance in depression outcomes. Plotting this interaction with the Excel software package showed a disordinal interaction (see Figure 1).

#### 3.1.2. Emotional Processing Coping Models for Depression Outcomes

In the second step of the model for emotional processing, IPV and emotional processing coping were added. Age (β = −0.145, *p* = 0.026), discrimination (β = 0.311, *p* = 0.000), IPV (β = 0.146, *p* = 0.039) and emotional processing (β = −0.210, *p* = 0.001) were significantly associated with depression outcomes. The addition of these variables accounted for an additional 5.7% of the variation in depression outcomes. In step three, the interaction term for IPV and emotional processing coping was added into the model. Age (β = −0.132, *p* = 0.042), discrimination (β = 0.307, *p* = 0.000), IPV (β = 0.145, *p* = 0.039), emotional processing (β = −0.397, *p* = 0.000), and the interaction variable (β = 0.232, *p* = 0.022) were significantly associated with depression outcomes. The addition of this variable accounted for an additional 2% of variation in depression outcomes. Plotting this interaction using the Excel software package showed a disordinal interaction (see Figure 2).

#### 3.1.3. Emotional Expression Coping Models for Depression Outcomes

In the second step of the model for emotional expression coping, IPV and emotional expression coping were added. Age (β = −0.140, *p* = 0.034), discrimination (β = 0.294, *p* = 0.000), IPV (β = 0.154, *p* = 0.032), and emotion expression coping (β = −0.170, *p* = 0.008) were significantly associated with depression outcomes. The addition of these variables accounted for an additional 4.2% of variation in depression outcomes. In step three, the interaction term for IPV and emotional expression coping was added into the model. The addition of this variable did not contribute significantly to depression outcomes. No significant changes in variation were found from step two to step three of this model.

#### 3.1.4. Social Support Coping Models for Depression Outcomes

In the second step of the model for social support coping model, IPV and social support were added. Age (β = −0.151, *p* = 0.023), discrimination (β = 0.307, *p* = 0.000), IPV (β = 0.145, *p* = 0.044), and social support coping (β = −0.146, *p* = 0.023) were significantly associated with depression outcomes. The addition of these variables accounts for an additional 3.5% of variation in depression outcomes. In step three, the interaction term for IPV and social support coping was added into the model. The addition of this variable did not contribute significantly to depression outcomes. No significant changes in variation were found from step two to step three of this model.

### 3.2. IPV and Coping Regression Models for Anxiety Outcomes

Four three-step regression analyses with anxiety as the dependent variable were conducted. All control variables were entered in at step one. In step two, IPV and each coping subscale were entered, and in step three, the interaction variable for IPV and each coping subscale were entered. All regression statistics are reported in Table 3. At step 1 for each model, control variables accounted for 16.6% of the variation in anxiety outcomes. While adjusting for all other variables, discrimination (β = 0.321, *p* = 0.000) was significantly associated with anxiety scores.

#### 3.2.1. Acceptance Coping Model for Anxiety Outcomes

In the second step of the model for acceptance coping, IPV and acceptance coping were added. IPV (β = 0.199, *p* = 0.007) was significantly associated with anxiety outcomes. The addition of these variables accounted for an additional 4.1% of variation in anxiety outcomes. In step three, the interaction term for IPV and acceptance coping was added into the model. The addition of this variable did not contribute significantly to anxiety outcomes. No significant changes in variation were found from step two to step three of this model.

#### 3.2.2. Emotional Processing Coping Model for Anxiety Outcomes

In the second step of the model for emotional processing coping, IPV and emotional processing were added. Race (β = 0.156, *p* = 0.038), discrimination (β = 0.242, *p* = 0.001), and IPV (β = 0.209, *p* = 0.005) were significantly associated with anxiety outcomes. The addition of these variables accounted for an additional 3.3% of variation in anxiety outcomes. In step three, the interaction term for IPV and emotional processing coping was added into the model. The addition of this variable did not contribute significantly to anxiety outcomes. No significant changes in variation were found from step two to step three of this model.

#### 3.2.3. Emotional Expression Coping Model for Anxiety Outcomes

In the second step of the model for emotional expression coping, IPV and emotional expression were added. Race (β = 0.163, *p* = 0.029), discrimination (β = 0.243, *p* = 0.001), and IPV (β = 0.204, *p* = 0.006) were significantly associated with anxiety outcomes. The addition of these variables accounted for an additional 3.7% of variation in anxiety outcomes. In step three, the interaction term for IPV and emotional expression coping was added into the model. The addition of this variable did not contribute significantly to anxiety outcomes. No significant changes in variation were found from step two to step three of this model.

#### 3.2.4. Social Support Coping Model for Anxiety Outcomes

In the second step of the model for social support coping, IPV and social support were added. Race (β = 0.160, *p* = 0.031), discrimination (β = 0.238, *p* = 0.001), and IPV (β = 0.208, *p* = 0.005) were significantly associated with anxiety outcomes. The addition of these variables accounted for an additional 3.6% of variation in anxiety outcomes. In step three, the interaction term for IPV and social support coping was added into the model. The addition of this variable did not contribute significantly to anxiety outcomes. No significant changes in variation were found from step two to step three of this model.

## 4. Discussion

The current study examined the relationship between intimate partner violence and mental health outcomes of depression and anxiety in a sample of transgender women. It was predicted that coping skills such as acceptance, emotional processing, emotional expression, and social support would moderate this relationship. As predicted, IPV was positively correlated with both anxiety and depression outcomes, suggesting that higher levels of IPV are associated with these increased negative psychological outcomes for transgender women, supporting current research [7,8,20]. In addition, across all of our models, experiences of discrimination were positively correlated with both anxiety and depression. Given the majority of participants in this study were transgender women of color, many of whom also identified as a sexual minority and reported living with HIV, we believe this finding emphasizes the continued need for researchers and clinicians to consider the negative psychological effects discrimination is evidenced to have on the everyday lives of transgender women, and even more so, transgender women who may be experiencing discrimination on a multi-level and intersectional basis [6,7,8,9,27,31,51].

Our findings did not support our prediction that coping skills would buffer the relationship of IPV and depressive symptoms and IPV and anxiety symptoms. However, for individuals with a history of less or no experience of IPV, who also reported a high use of emotional processing and acceptance coping, we did observe a buffering effect on depression symptoms specifically. Social support coping was also negatively associated with depression symptoms. These findings provide evidence that, regardless of IPV history, for transgender women living with depressive symptoms, the use of adaptive coping strategies such as emotional processing, acceptance, and social support may provide protective effects.

Evidence-based interventions (EBIs) for transgender women may benefit from these findings. Specifically, for transgender women living with depression, interventions including a social support component (e.g., group-based), and emphasis on emotional processing and acceptance, may be of particular importance. Recent research has emphasized the development and use of TGD-affirmative dialectical behavioral therapy skills training groups [52]. However, there is limited research looking at dialectical behavioral therapy with transgender women, specifically. In general, there is a gap in the research assessing whether standard EBIs are feasible, acceptable, and effective for this population. 

Most EBIs have been developed for and tested with cisgender populations, thus there is a need for interventions and evidence-based practices tailored to the unique needs of transgender women [53]. Future intervention research should consider a community engaged research (CEnR) approach to intervention development and adaptation. In this approach, community partners and members are actively engaged in the research process, are treated as equal partners to the researchers throughout, and share in the decision-making [54]. Particularly, this approach may be useful in addressing health disparities, increasing impact of the research being conducted, and increasing the uptake of interventions and evidence-based practices among disadvantaged populations [54,55,56]. 

Lastly, forms of structural and systemic discrimination, such as legislation aimed at restricting and banning access to gender-affirming care for TGD populations has increased in the past year [57]. Not only does discrimination such as this decrease access to care, but it is evidenced to increase barriers to healthcare such as deterring transgender and gender-diverse people from seeking healthcare [58]. We recommend two advances. First, healthcare practitioners would benefit from receiving education and training that is culturally relevant, and inclusive of the healthcare needs and experiences of TGD individuals. Lastly, policies protecting gender-affirming and TGD-inclusive healthcare, at all levels, are needed to establish lasting institutional changes that ensure access to and the receival of quality care that is informed by TGD-specific experiences and needs. 

This study was not without limitation. The data were secondary, from a previous study, limiting the scope of decision making for the current researchers regarding measure selection and participant data. Additionally, the data utilized were cross-sectional; assessing these relationships using longitudinal data would provide support for the hypothesized temporal order of these associations. Also, the coping measure utilized in this study was developed using items from multiple coping scales. This measure has yet to undergo psychometric testing, as such, our future research aims to validate its use for this population. Finally, this sample should not be considered representative of the general population of transgender women, as participants in this sample were all living in NYC and were recruited for a waitlist control trial of an intervention targeting HIV risk and substance use. Given that the primary study used targeted ads and outreach to recruit transgender women eligible for this intervention study, our sample may reflect elevated numbers for HIV prevalence and HIV risk behavior, including history of substance use and sex work, as compared to the general population of transgender women.

## 5. Conclusions

Current research suggests that intimate partner violence (IPV) is associated with negative mental health outcomes for transgender women [9]. The current study aimed to examine the relationship between intimate partner violence, coping skills, depression, and anxiety symptoms in a sample of transgender women. Hierarchical regression analyses were conducted examining the relationship of IPV and depression and anxiety symptoms, where coping skills moderate this relationship. The findings of this current study show adaptive coping to have potential protective effects for negative mental health outcomes such as a depression among transgender women. Further research is recommended to assess which forms of coping may be most helpful in mitigating the negative effects of IPV and discrimination [32,33,34,35,36] on mental health outcomes, specifically among racially diverse, sexual minority-identifying, and HIV-positive transgender women. Further research examining culturally informed and relevant, evidence-based, socially supportive interventions for coping with negative psychological outcomes for transgender women in relation to IPV and other forms of violence is needed. It is the researchers hope that this study further contributes to more knowledge, prevention, and suggestions for the use of adaptive behavioral and TGD-inclusive healthcare strategies, for positive health outcomes and evidence-based healthcare science and healthcare practice for transgender women.

## Figures and Tables

**Figure 1 ijerph-20-05927-f001:**
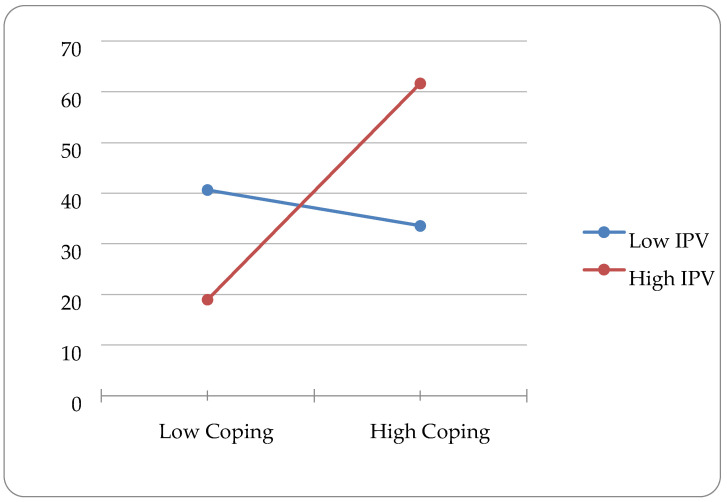
Interaction Effect of Acceptance Coping and IPV Predicting Depression.

**Figure 2 ijerph-20-05927-f002:**
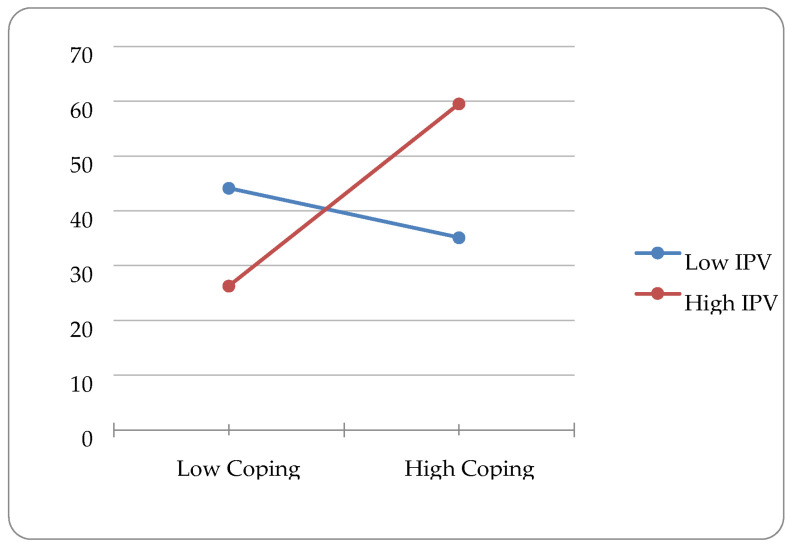
Interaction Effect of Emotional Processing Coping and IPV Predicting Depression.

**Table 1 ijerph-20-05927-t001:** Sample Demographics.

	N (%)	Mean (SD)
Age		34.27 (11.69)
Race/Ethnicity		
Black	67 (31.6%)	
Latino	60 (28.3%)	
White	52 (24.5%)	
Other	33 (15.6%)	
Income (USD)		
Less than 20k	170 (80.2%)	
20k to 49k	35 (16.5%)	
50k or more	7 (3.3%)	
Employment		
Employed	56 (26.4%)	
Unemployed	156 (73.6%)	
Education		
High School	88 (41.5%)	
Some College	62 (29.2%)	
4-Year Degree	49 (23.1%)	
Graduate School	13 (6.1%)	
Sexual Identity		
Heterosexual	101 (47.6%)	
Bisexual	39 (18.4%)	
Queer	31 (14.6%)	
Gay	29 (13.7%)	
Lesbian	12 (5.6%)	
HIV +	73 (34.4%)	
Relationship Status		
Single	108 (50.9%)	
In a Relationship	104 (49.1%)	
Sex Work	136 (64.2%)	
Substance Use	148 (69.8%)	
IPV	138 (65.1%)	
BSI-Anxiety		1.04 (1.05)
Depression		20.85 (11.42)
Discrimination		20.34 (14.34)

**Table 2 ijerph-20-05927-t002:** Bi-Variate Correlations among Measures.

	1	2	3	4	5	6	7
1. Number IPV Experiences	-	0.114	0.042	0.064	0.038	0.221 **	0.261 **
2. COPE-Acceptance		-	0.555 **	0.399 **	0.482 **	−0.003	0.131
3. COPE-Emotional Processing			-	0.749 **	0.685 **	−0.196 **	0.005
4. COPE-Emotional Expression				-	0.682 **	−0.171 *	0.054
5. COPE-Social Support					-	−0.132	0.057
6. CES-D						-	0.650 **
7. BSI-ANX							-
Mean (SD)	3.63 (4.03)	11.78 (3.12)	12.44 (2.99)	12.00 (3.36)	12.07 (3.05)	20.85 (11.42)	1.04 (1.05)
α	0.92	0.78	0.81	0.87	0.82	0.80	0.91

* *p* < 0.05, ** *p* < 0.001.

**Table 3 ijerph-20-05927-t003:** Hierarchical regression analyses examining the moderating effect of four coping strategies.

	Models for Depression									Models for Anxiety								
Coping Skill/Step	R2	Step 1 β	*p*	R2	Step 2 β	*p*	R2	Step 3 β	*p*	R2	Step1 β	*p*	R2	Step 2 β	*p*	R2	Step 3 β	*p*
Moderating Effect of Acceptance	0.199			0.217			0.243			0.166			0.208			0.209		
1 Age		−0.155	0.022		−0.159	0.018		−0.146	0.028		−0.102	0.136		−0.100	0.135		−0.097	0.150
Race (White vs. other)		0.109	0.135		0.122	0.096		0.118	0.104		0.129	0.084		0.162	0.029		0.161	0.030
Income		0.063	0.362		0.074	0.283		0.068	0.316		0.096	0.172		0.130	0.063		0.128	0.066
College		−0.056	0.429		−0.070	0.320		−0.081	0.244		−0.085	0.238		−0.090	0.208		−0.092	0.197
Sexual Identity/Ref.(Heterosexual)		0.038	0.581		0.024	0.732		0.035	0.607		0.033	0.635		0.028	0.681		0.031	0.654
Everyday Discrimination		0.350	0.000		0.300	0.000		0.293	0.000		0.321	0.000		0.237	0.001		0.235	0.002
HIV+		−0.033	0.619		−0.059	0.378		−0.052	0.433		0.010	0.876		−0.019	0.777		−0.017	0.797
In a Relationship		0.041	0.529		0.048	0.460		0.053	0.402		0.033	0.618		0.037	0.568		0.038	0.556
Sex Work		0.004	0.952		−0.024	0.731		−0.031	0.657		−0.024	0.731		−0.061	0.393		−0.062	0.382
2 IPV		-	-		0.146	0.046		0.143	0.047		-	-		0.199	0.007		0.198	0.007
Acceptance		-	-		−0.054	0.402		−0.262	0.010		-	-		0.093	0.151		0.045	0.666
3 IPV × Acceptance		-	-		-	-		0.266	0.009		-	-		-	-		0.062	0.547
Moderating Effect of Emotional Processing	0.199			0.256			0.275			0.166			0.199			0.202		
1 Age		−0.155	0.022		−0.145	0.026		−0.132	0.042		−0.102	0.136		−0.105	0.119		−0.100	0.140
Race (White vs. other)		0.109	0.135		0.097	0.179		0.068	0.346		0.129	0.084		0.156	0.038		0.145	0.057
Income		0.063	0.362		0.061	0.364		0.051	0.448		0.096	0.172		0.121	0.083		0.117	0.095
College		−0.056	0.429		−0.073	0.289		−0.062	0.364		−0.085	0.238		−0.099	0.167		−0.094	0.187
Sexual Identity/Ref.(Heterosexual)		0.038	0.581		0.031	0.642		0.040	0.547		0.033	0.635		0.019	0.778		0.023	0.742
Everyday Discrimination		0.350	0.000		0.311	0.000		0.307	0.000		0.321	0.000		0.242	0.001		0.240	0.001
HIV +		−0.033	0.619		−0.079	0.225		−0.099	0.130		0.010	0.876		−0.024	0.726		−0.031	0.647
In a Relationship		0.041	0.529		0.060	0.337		0.071	0.253		0.033	0.618		0.040	0.539		0.044	0.499
Sex Work		0.004	0.952		−0.031	0.651		−0.034	0.622		−0.024	0.731		−0.064	0.371		−0.065	0.364
2 IPV		-	-		0.146	0.039		0.145	0.039		-	-		0.209	0.005		0.209	0.005
Emotional Processing		-	-		−0.210	0.001		−0.397	0.000		-	-		0.004	0.953		−0.068	0.524
3 IPV × Emotional Processing		-	-		-	-		0.232	0.022		-	-		-	-		0.089	0.398
Moderating Effect of Emotional Expression	0.199			0.241			0.244			0.166			0.203			0.205		
1 Age		−0.155	0.022		−0.140	0.034		−0.134	0.044		−0.102	0.136		−0.111	0.101		−0.106	0.119
Race (White vs. other)		0.109	0.135		0.104	0.151		0.098	0.179		0.129	0.084		0.163	0.029		0.158	0.035
Income		0.063	0.362		0.078	0.249		0.071	0.301		0.096	0.172		0.121	0.081		0.116	0.099
College		−0.056	0.429		−0.081	0.247		−0.081	0.246		−0.085	0.238		−0.093	0.192		−0.093	0.192
Sexual Identity/Ref.(Heterosexual)		0.038	0.581		0.008	0.904		0.015	0.821		0.033	0.635		0.027	0.699		0.032	0.644
Everyday Discrimination		0.350	0.000		0.294	0.000		0.299	0.000		0.321	0.000		0.243	0.001		0.247	0.001
HIV+		−0.033	0.619		−0.070	0.288		−0.075	0.258		0.010	0.876		−0.019	0.777		−0.023	0.738
In a Relationship		0.041	0.529		0.059	0.353		0.065	0.309		0.033	0.618		0.036	0.586		0.040	0.542
Sex Work		0.004	0.952		−0.024	0.734		−0.026	0.711		−0.024	0.731		−0.063	0.372		−0.065	0.361
2 IPV		-	-		0.154	0.032		0.150	0.037		-	-		0.204	0.006		0.201	0.007
Emotional Expression		-	-		−0.170	0.008		−0.242	0.017		-	-		0.058	0.372		0.007	0.943
3 IPV × Emotional Expression		-	-		-	-		0.092	0.356		-	-		-	-		0.069	0.503
Moderating Effect of Social Support	0.199			0.234			0.243			0.166			0.202			0.205		
1 Age		−0.155	0.022		−0.151	0.023		−0.145	0.028		−0.102	0.136		−0.107	0.112		−0.104	0.125
Race (White vs. other)		0.109	0.135		0.112	0.123		0.102	0.163		0.129	0.084		0.160	0.031		0.154	0.039
Income		0.063	0.362		0.062	0.365		0.060	0.377		0.096	0.172		0.128	0.069		0.127	0.071
College		−0.056	0.429		−0.072	0.304		−0.065	0.349		−0.085	0.238		−0.096	0.177		−0.092	0.195
Sexual Identity/Ref.(Heterosexual)		0.038	0.581		0.013	0.844		0.027	0.695		0.033	0.635		0.025	0.715		0.033	0.637
Everyday Discrimination		0.350	0.000		0.307	0.000		0.312	0.000		0.321	0.000		0.238	0.001		0.242	0.001
HIV+		−0.033	0.619		−0.069	0.295		−0.075	0.259		0.010	−0.156		−0.019	0.778		−0.022	0.744
In a Relationship		0.041	0.529		0.053	0.408		0.065	0.312		0.033	0.618		0.038	0.563		0.044	0.498
Sex Work		0.004	0.952		−0.023	0.737		−0.030	0.662		−0.024	0.731		−0.063	0.372		−0.067	0.344
2 IPV		-	-		0.145	0.044		−0.225	0.370		-	-		0.208	0.005		−0.004	0.986
Social Support		-	-		−0.146	0.023		−0.230	0.006		-	-		0.056	0.391		0.007	0.933
3 IPV × Social Support		-	-		-	-		0.393	0.125		-	-		-	-		0.226	0.389

## Data Availability

Due to confidentiality, the datasets utilized in the current work are not publicly accessible. Data sharing is not applicable to this article.

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
