# Peer review of "Coping Strategies as a Moderator for the Association between Intimate Partner Violence and Depression and Anxiety Symptoms among Transgender Women"

_ijerph, 2023, doi:10.3390/ijerph20115927_

Round 1

Reviewer 1 Report

Thank you for the opportunity to review the manuscript. Overall, a current topic for a broader readership and further exploration of this topic to test the interaction between intimate partner violence, and to know if coping skills moderate the association between IPV history and anxious or depressive symptomatology in U.S.

A few questions / comments and suggestions:

In Line 373-374, the overall content requires detail elaboration, relevant to the study is not clear.

In Line 377-378, to elaborate detailly for “negative mental outcomes for this study, relevant to the study is not clear.

In Line 378-379, how to elaborate “discrimination was positively”, relevant to the study is not clear.

In Line 391-393, be more detailly describe the meaning of “how to decrease discrimination and increase support for transgender women”, relevant to the study is not clear.

In Line 393-395, how to this study analyze the result as “not only with negative mental health outcomes associated with IPV”, relevant to the study is not clear.

In 358-397, suggest expanding the content for this discussion part to share your perspectives for this study, relevant to the study is not clear.

There should be a separate section in the discussion section, stating the significance and implications of this study to the international nursing practice. The section should be specific and based on the findings.

Thank you for the opportunity to review the manuscript. Overall, a current topic for a broader readership and further exploration of this topic to test the interaction between intimate partner violence, and to know if coping skills moderate the association between IPV history and anxious or depressive symptomatology in U.S.

A few questions / comments and suggestions:

In Line 373-374, the overall content requires detail elaboration, relevant to the study is not clear.

In Line 377-378, to elaborate detailly for “negative mental outcomes for this study, relevant to the study is not clear.

In Line 378-379, how to elaborate “discrimination was positively”, relevant to the study is not clear.

In Line 391-393, be more detailly describe the meaning of “how to decrease discrimination and increase support for transgender women”, relevant to the study is not clear.

In Line 393-395, how to this study analyze the result as “not only with negative mental health outcomes associated with IPV”, relevant to the study is not clear.

In 358-397, suggest expanding the content for this discussion part to share your perspectives for this study, relevant to the study is not clear.

There should be a separate section in the discussion section, stating the significance and implications of this study to the international nursing practice. The section should be specific and based on the findings.

Reviewer 2 Report

Thank you for giving me the opportunity to review this study.  It is clear that this is an area of ongoing interest and relevance and the introduction provides a clear perspective on the current state of play.  However this is a very comprehensive literature already and what is a lot less clear is how the current study adds something beyond what is already known.  From what is presented, the specific risks to and impact on transgender women as well as the potential for support and coping are already established.  So the introduction and research aims for me need to be much more definitive about how this study adds value to the area.  

This would also assist with the results.  The use of hierarchical linear regression would imply a clear analytic strategy and model which justifies a specific order of entry of the variables.  For me this is implied rather than stated and, although the results are logically presented and the analysis overall is well conducted, this felt at times like fishing rather an a purposeful a priori strategy. This is something which needs to be more clearly articulated. There was evidently a strategy because steps were articulated but it was not clear how this strategy was a model built on extending/confirming the previous research - including which parts were extension and which were confirmation.  Finally this also needed to be more clearly articulated in the discussion.

In terms of the conduct of the study itself, I felt that the use of large scale secondary data was both efficient and effective as an approach to explore the issue, that the choice of measures were appropriate in terms of their reliability and validity and that the choice of control variables was comprehensive - allowing for more nuanced comparisons to be made.  The statistical analysis was well carried out and reported and, if the model were clearer would be entirely appropriate. The article as a whole was very well written and there is a clear and progressive direction of travel. If the authors could however put themselves into the position of an external reader who does not know what the aims are and then amend the argument throughout to make the novelty and contribution more explicit, that would significantly improve the work. My feeling is that the authors were simply "too involved" in the process of producing the article and just need to take a step back and remind themselves that the readership would be less familiar.  So this should be an easy amendment to achieve.  

I had two minor points to improve clarity.  First there seems to be a transposition of irrelevant information in lines 136-138 at the end of the introduction. Second, there needs to be a clearer operationalisation of the difference between transgender women and women with transgender experience in the participants section.  Are the latter group women who are not transgender but who know or have involvement with those who are? This needed to be clearer.  Otherwise I did not spot any concerns with the writing or presentation.
